# Efficacy and Safety of Low-Dose Cyclosporine Relative to Immunomodulatory Drugs Used in Atopic Dermatitis: A Systematic Review and Meta-Analysis

**DOI:** 10.3390/jcm12041390

**Published:** 2023-02-09

**Authors:** Kyunghoon Kim, Mina Kim, EunHee Rhee, Mi-Hee Lee, Hyeon-Jong Yang, Suyeon Park, Hwan Soo Kim

**Affiliations:** 1Department of Pediatrics, Seoul National University College of Medicine, Seoul 03080, Republic of Korea; 2Department of Applied Statistics, Chung-Ang University, Seoul 06974, Republic of Korea; 3SCH Biomedical Informatics Research Unit, Soonchunhyang University Seoul Hospital, Seoul 04401, Republic of Korea; 4Department of Pediatrics, Soonchunhyang University Seoul Hospital, Soonchunhyang University College of Medicine, Seoul 04401, Republic of Korea; 5Department of Pediatrics, Incheon Medical Center, Incheon 22532, Republic of Korea; 6Department of Biostatistics, Soonchunhyang University Seoul Hospital, Seoul 04401, Republic of Korea; 7Department of Pediatrics, College of Medicine, The Catholic University of Korea, Seoul 06591, Republic of Korea

**Keywords:** atopic dermatitis, cyclosporine A, efficacy

## Abstract

Cyclosporine A (CsA) is effective in treating moderate-to-severe atopic dermatitis (AD). This systematic review and meta-analysis aimed to summarize the effectiveness and safety of low-dose (<4 mg/kg) versus high-dose (≥4 mg/kg) CsA and other systemic immunomodulatory agents in patients with AD. Five randomized controlled trials met the inclusion criteria. The meta-analysis included 159 patients with moderate-to-severe AD who were randomized to receive low-dose CsA, and 165 patients randomized to receive high-dose CsA and other systemic immunomodulatory agents. We found that low-dose CsA was not inferior to high-dose CsA and other systemic immunomodulatory agents in reducing AD symptoms [standard mean difference (SMD) −1.62, 95% confidence interval (CI) −6.47; 3.23]. High-dose CsA and other systemic immunomodulatory agents showed a significantly lower incidence of adverse events [incidence rate ratio (IRR) 0.72, 95% CI 0.56; 0.93], however, after sensitivity analysis, there was no difference between the two groups except for one study (IRR 0.76, 95% CI 0.54; 1.07). Regarding serious adverse events requiring discontinuation of treatment, we observed no significant differences between low-dose CsA and other systemic immunomodulatory agents (IRR 1.83, 95% CI 0.62; 5.41). Our study may justify the use of low-dose CsA rather than high-dose CsA and other systemic immunomodulatory agents in moderate-to-severe AD.

## 1. Introduction

Atopic dermatitis (AD) is the most common, chronic inflammatory skin condition, affecting 5–8% of adults and 11–20% of children [1,2,3]. Patients with AD present with mild local to severe systemic symptoms, such as itching, pain, and sleep disturbances, leading to a substantially lower quality of life [1,2]. Approximately one-third of children and half of adults with AD have a moderate or severe form of the disease, thus, requiring systemic therapy [4].

Systemic immunomodulatory agents used to treat AD include cyclosporine A (CsA), methotrexate (MTX), azathioprine, mycophenolate mofetil (MMF), and monoclonal antibodies, such as dupilumab and Janus kinase inhibitors [5,6,7]. Specifically, CsA is known to be effective in AD, but side effects have been reported, such as hypertension and nephrotoxicity [8,9]. Reduction in CsA doses has been suggested as an optimal way to allow long-term CsA with reduced adverse effects (AEs) [10].

The aim of this systematic review and meta-analysis of randomized clinical trials (RCTs) is to analyze the efficacy and safety according to the dose of CsA in moderate-to-severe AD.

## 2. Materials and Methods

### 2.1. Literature Search Strategy

We performed a systematic review and meta-analysis in accordance with the Preferred Reporting Items for Systematic Reviews and Meta-Analyses (PRISMA) guidelines [11]. The population-intervention-comparison-outcome question used for our search strategy was as follows: “Is CsA more effective than the other drugs in patients with AD? Is CsA safer than the other drugs? What is the most effective and safe dose of CsA for patients with AD?”

We performed a systematic search using a protocol with five electronic databases, namely PubMed, Embase, Cochrane Library, Clinical Trial Registry, and the World Health Organization International Clinical Trials Registry Platform. Here, RCTs comparing CsA with other interventions in patients with AD were eligible for inclusion. We used the search terms listed in the Appendix A to search the electronic databases. (Appendix A). We included studies published until 2 July 2021, and imposed no language or publication restrictions.

### 2.2. Study Selection

Two reviewers (K.K. and H.S.K.) independently evaluated the titles and abstracts obtained from the first screening. Articles that did not focus on CsA use in AD as well as review articles were excluded from this initial screening. After this, the reviewers independently reviewed the full texts of the remaining articles to determine whether they met the following eligibility criteria: (1) RCT comparing the efficacy and safety of CsA in patients with AD and (2) comparisons of outcome measures, including clinical severity, quality of life, and AEs. The primary outcome was the relief of AD symptoms, quantitatively measured using validated scoring systems. The secondary outcome was the occurrence of AEs. Review articles, abstracts without full-text publications, and case study reports were excluded. Disagreements between the reviewers in the selection of particular studies were resolved after discussion with a third reviewer (H.J.Y.).

### 2.3. Data Extraction and Quality Assessment

Both reviewers (K.K. and H.S.K.) extracted the data from each eligible study using a structured procedure. Data could be classified by the sample characteristics, the intervention details, and the measurement of outcomes. Outcome measures were divided into primary outcomes, which assessed the efficacy of CsA, and secondary outcomes, assessing the safety of CsA. The reviewers independently assessed the risk of bias for each study using the criteria outlined in the *Cochrane Handbook for Systematic Reviews of Interventions* [12]. Disagreements between the reviewers in the selection of particular studies were settled after discussion with a third reviewer (H.J.Y.).

### 2.4. Data Analysis

Statistical analyses were performed using R software (R Foundation for Statistical Computing, Vienna, Austria) version 4.2.0, “meta” package. To identify differences in treatment effects and AEs between groups, the pooled estimates used standard mean difference (SMD) and incidence relative risk (IRR), respectively, and a random effect model was used with the DerSimonian–Laird method. All tests were two-tailed, and a *p*-value < 0.05 was deemed statistically significant. We used the I² statistic to assess heterogeneity in the results of individual studies (I^2^ >50% was used as a threshold to indicate significant heterogeneity).

In the treatment effect analysis, the mean difference between baseline and the last follow-up time point for each group in the study was calculated using descriptive statistics for meta-analysis, and measurement bias between studies was corrected using SMD as summary statistics. Concerning the analysis of AEs, the formula for incidence rate (IR, person-AE) for each group was calculated as follows:$$\mathrm{IR}=\frac{\mathrm{Total}\;\mathrm{number}\;\mathrm{of}\;\mathrm{AEs}}{\left(\mathrm{Total}\;\mathrm{number}\;\mathrm{of}\;\mathrm{participants}\;\mathrm{in}\;\mathrm{the}\;\mathrm{group}\right)\times \left(\mathrm{The}\;\mathrm{number}\;\mathrm{of}\;\mathrm{AEs}\right)}$$
where AEs were any undesirable symptoms associated with the use of a medication in the patients in each RCT study. Serious AEs were identified as such when they caused the discontinuation of the RCT. The IR was also calculated in these cases.

In this study, low-dose CsA (<4 mg/kg) was used as the control group, and high-dose CsA (≥4 mg/kg), MTX, MMF, and prednisolone (PRD) were used as the experimental groups. 

## 3. Results

### 3.1. Study Selection

A total of 255 citations were initially screened on the databases, and 146 individual publications were identified. Of these, 120 studies were excluded after reviewing the titles and abstracts, leaving 26 articles for full-text review. Finally, five articles were selected for eligibility and were included in our final meta-analysis (Figure 1) [13,14,15,16,17].

### 3.2. Study Characteristics

Overall, 324 patients were included in our meta-analysis. Of this number, 159 patients were administered low-dose CsA (control group), and 165 patients were administered high-dose CsA and other systemic immunomodulatory agents (experimental groups). The CsA treatment was initiated at 2.5–4 mg/kg/day in the control group [13,16,17]. In one study, CsA increased in patients with poor response 8 weeks later, and the period after 8 weeks of dose increase was defined separately as a low-dose extended group [13]. In another study, CsA administration was initiated at 150 mg/day in adults [14]. In the experimental group, CsA treatment was initiated at 300 mg/day [14]. In one study, CsA treatment started at 5 mg/kg/day, but was reduced at an early stage. [15] The period of 0–6 weeks using 5 mg/kg/day was defined as the experimental group, and the period after 6 weeks when the dose of the drug was reduced to 3 mg/kg/day was divided into the control group. Additionally, the period after 6 weeks when MMF administration was at 1440 mg/day was defined as the experimental group. Two studies started with MTX administration at 7.5–15 mg/week, and another started with PRD administration at 0.5–0.8 mg/day, and these were defined as the experimental group [13,16,17]. In most studies, SCORing Atopic Dermatitis (SCORAD) scores were used to evaluate the improvement of AD symptoms in patients from 3–24 weeks after treatment. In one study, the total body surface area (TBSA) score was assessed 8 weeks after drug administration (Table 1) [14]. The risk of bias in the included studies was evaluated, as shown in Figure 2.

### 3.3. Outcome Measures

A summary of the data from the studies included in the meta-analysis is presented in Table 2. Figure 3 shows the overall SMD and 95% confidence interval (CI) for each intervention compared to low-dose CsA. We found that low-dose CsA was not inferior to high-dose CsA and other systemic immuno-modulatory agents in reducing AD symptoms (SMD = −1.62, 95% CI −6.47; 3.23). 

The overall incidence rate ratio (IRR) of AEs and 95% CI are shown in Figure 4a. The IRR was significantly lower in the high-dose CsA and other systemic immunomodulatory agents group (IRR 0.72, 95% CI 0.56; 0.93). Haeck’s study has a relatively long observation period compared to other studies. Therefore, milder adverse events were reported. After the sensitivity analysis (Appendix A), except for Haeck’s study, there was no significant difference between the two groups when they were analyzed (IRR 0.76, 95% CI 0.54; 1.07) (Figure 4b). Regarding serious AEs requiring discontinuation of treatment, the IRR in the low-dose CsA group was not significantly different from other interventions, including high-dose CsA and other systemic immunomodulatory agents (IRR 1.83, 95% CI 0.62; 5.41) (Figure 4c). When comparing low-dose CsA with high-dose CsA, low-dose CsA showed no significant difference in incidence of AEs and serious AEs. 

## 4. Discussion

Using this systematic review and meta-analytical approach, we compared the efficacy and AEs of low-dose CsA with high-dose CsA and various systemic immunomodulators in AD. This meta-analysis comprised 5 RCTs, which included a total of 324 patients. Analyses of outcomes in patients receiving various systemic immunomodulators showed no significant differences in severity score improvement compared to patients receiving low-dose CsA. Low-dose CsA showed a higher incidence of AEs than other systemic immunomodulators. However, after sensitivity analysis, there was no difference between the low-dose CsA and other systemic immunomodulators in the occurrence of AEs. Furthermore, there was no difference between low-dose CsA and other systemic immunomodulators in the occurrence of serious AEs requiring discontinuation of the clinical trial.

Typically, CsA is considered to be the first-line option for patients with severe AD who require systemic immunosuppressive treatment [18]. A previous meta-analysis and review demonstrated the efficacy of CsA in AD with a 55% improvement on average after 6–8 weeks of treatment [19]. In general, it is recommended to start with a higher dose of 4–5 mg/kg/day to obtain a good initial result unless the patient is old or suffers from relevant concomitant diseases [20]. Although CsA was more effective than a placebo, all scores returned to pre-treatment values 8 weeks after the cessation of CsA therapy in most patients [19]. We found that the use of low-dose CsA showed non-inferior efficacy compared to high-dose CsA and other systemic immunomodulators, which might favor the initial use of low-doses of CsA. However, we also observed a significantly higher incidence of AEs with low-dose CsA than with other systemic immunomodulators. However, after sensitivity analysis, there was no difference between the two groups except for one study, and when comparing low-dose CsA with high-dose CsA, low-dose CsA showed no significant difference in incidence of AEs and serious AEs. Patients receiving CsA should be monitored for hypertension and renal toxicity, because CsA is known to induce structural and organic kidney damage. Nephrotoxic effects are more likely to occur if the daily CsA dose exceeds 5 mg/kg body weight and serum creatinine levels are elevated, or in older patients; however, these effects are not related to treatment duration [18,20]. One study supported such finding in which the mean age of the twenty-two patients with a clinically relevant serum creatinine increase was significantly higher than the mean age of patients without this increase in serum creatinine [20]. Previous studies showed that the cumulative incidence rates of AEs ranged from 0% to 13% for CsA and withdrawal rates ranged from 0% to 63% for CsA [21,22,23]. Our results showed that the incidence of serious AEs was not significantly different between low-dose CsA and other systemic immunomodulators.

We found no significant differences between the efficacies of low-dose CsA and MTX. Two previous trials compared MTX and CsA treatments, one in children and one in adults [13,16]. The trial performed on children showed a significant improvement in SCORAD after 12 weeks of both treatments [16]. The trial in adults concluded that MTX (15 mg/week) was inferior to low-dose CsA (2.5 mg/kg/day) regarding SCORAD reduction after 12 weeks [13]. We observed no significant difference in the incidence rate of AEs and serious AEs between the CsA and MTX groups. A previous study reported that the number of treatment-related AEs was significantly higher in the CsA group than in the MTX group [13]. However, none of the reported adverse reactions resulted in discontinuing or decreasing the drug dose [16]. Infections, gastrointestinal disturbances, and in rare cases, myelotoxicity, are AEs that may limit the use of MTX [13,19,24,25]. Because MTX is hepatotoxic and teratogenic, women of childbearing potential must use effective contraception during therapy [26].

We found a significant improvement in the severity of AD, with a lower incidence of AEs after treatment with MMF compared with low-dose CsA. The incidence of serious AEs did not differ significantly between the groups. One trial with a high risk of bias showed equal efficacy of CsA and MMF in adults with SCORAD at 12 weeks, but MMF showed a more delayed response [15]. Nausea and diarrhea were the most relevant gastrointestinal AEs of MMF. Side effects were most common upon treatment initiation and tended to improve over time. Recent data indicate that MMF should be discontinued 6 weeks before a planned pregnancy [27]. Future studies with a low risk of bias are needed to accurately determine the increased effectiveness of MMF over CsA.

No significant differences were observed between the efficacy of low-dose CsA and PRD. A previous study demonstrated that CsA was superior to oral PRD in achieving stable remission, with no relapse within the 12-week follow-up [17]. However, this trial was stopped due to safety issues in the PRD group (high relapse rates); 52% of patients receiving PRD and 29% receiving CsA withdrew from the trial due to AEs [17]. Therefore, an AD treatment guideline recommended that while short-term treatment with oral glucocorticosteroids was moderately effective, systemic steroids have a largely unfavorable risk/benefit ratio for the treatment of AD [18].

This study had some limitations. There was some heterogeneity in the design of the included trials. In particular, the use of background therapy (topical anti-inflammatory medications) varied between studies. Second, this meta-analysis included a few RCTs with a small number of patients with a large variation in dosage of drugs and treatment duration, which should be interpreted with care. Thirdly, CsA is licensed to be used in children over 24 months of age in Korea. Since this study included children over 8 years of age, our recommendations may not be applicable in children between 24 months and 8 years of age. Moreover, there are no comprehensive long-term safety trials for more than 1 year for any treatment. 

## 5. Conclusions

In conclusion, low-dose CsA showed non-inferior efficacy compared to high-dose CsA and various systemic immunomodulators. Further studies, including planned RCTs, will help to confirm and improve the accuracy of our obtained results and provide estimates for children in terms of long-term outcomes and side effects.

## Figures and Tables

**Figure 1 jcm-12-01390-f001:**
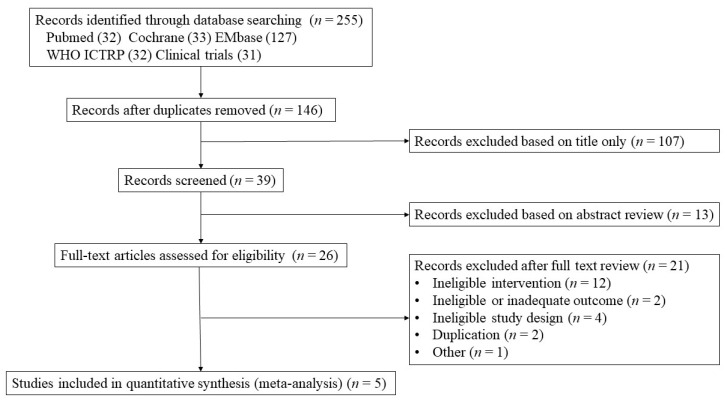
PRISMA flow diagram.

**Figure 2 jcm-12-01390-f002:**
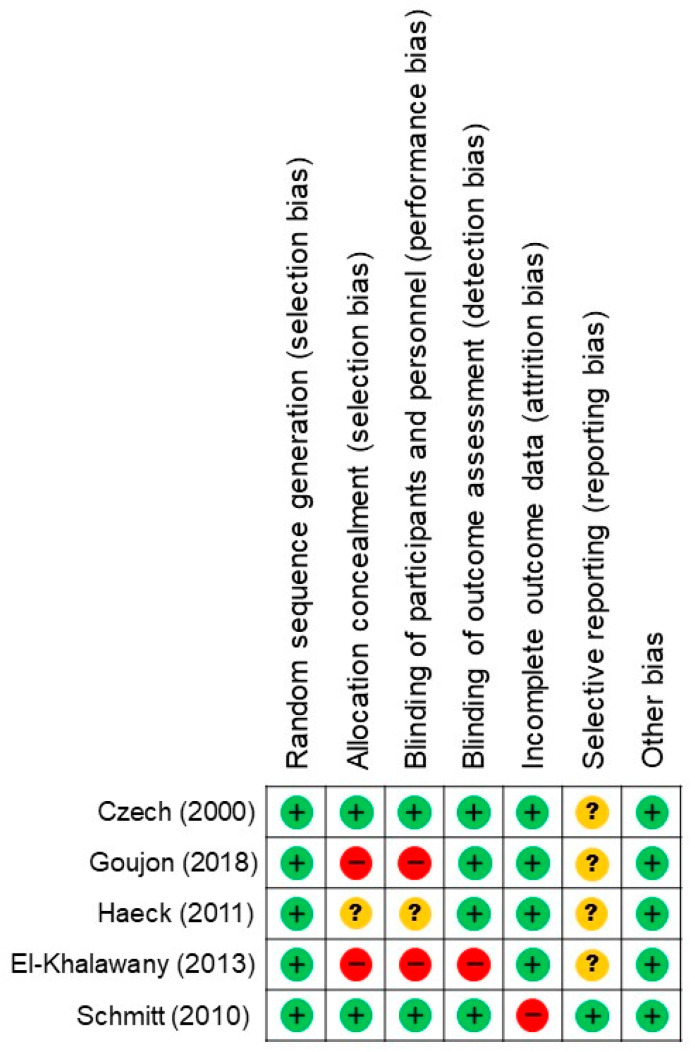
Risk of bias assessment of the randomized controlled trials included in the meta-analysis [13,14,15,16,17]. The symbols “+”, “−”, and “?” indicate low, high, and unclear risk of bias, respectively.

**Figure 3 jcm-12-01390-f003:**
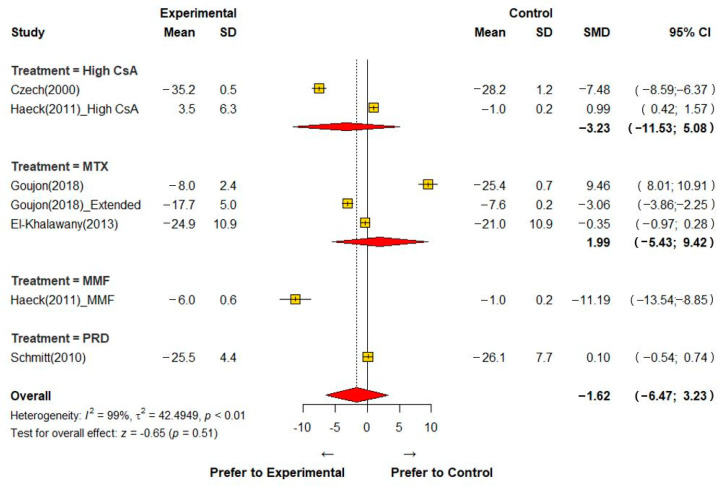
Meta-analysis of the effects on reducing atopic dermatitis symptoms [13,14,15,16,17]. CI; confidence interval, CsA; Cyclosporine A, MMF; mycophenolate mofetil, MTX; methotrexate, PRD; prednisolone, SD; standard deviation, SMD; standard mean difference.

**Figure 4 jcm-12-01390-f004:**
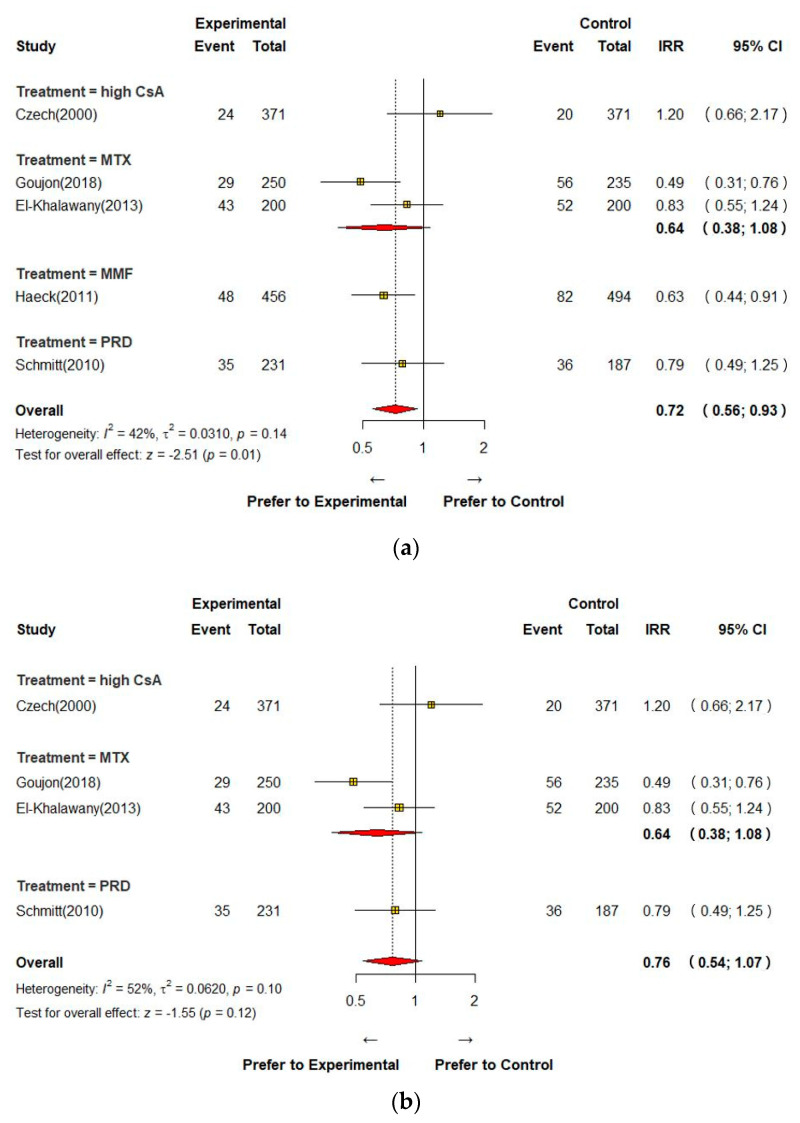
Meta-analysis of the adverse events (**a**), the adverse events excluding Heack’s study (**b**), and the serious adverse events (**c**) [13,14,15,16,17]. CI; confidence interval, CsA; Cyclosporine A, IRR; incidence relative risk, MMF; mycophenolate mofetil, MTX; methotrexate, PRD; prednisolone.

**Table 1 jcm-12-01390-t001:** Characteristics of studies included in the meta-analysis.

Study	Characteristics					Control Group		Experimental Group		Considered Complications
	Country	STUDY PERIOD	Design	Population	AD Severity	*n*	Intervention	*n*	Intervention	
Czech et al. (2000) [14]	Germany	Not described	RCT	Adults (≥18 years)	Severe	53	Start CsA 150 mg/day for 2 weeks and 50% reduced according to clinical response	50	Start CsA 300 mg/day for 2 weeks and 50% reduced according to clinical response	Skin disease, pain/nervous system disorder, GI disorder, metabolic disorder, cardiovascular disorder, gingival bleeding, others
Schmitt et al. (2010) [17]	Germany	February 2007 to November 2008	RCT	Adults (18–55 years)	Severe	17	CsA 2.7–4.0 mg/kg/day for 6 weeks	21	Start prednisolone 0.5–0.8 mg/kg/d and tapered within 2 weeks	Exacerbation/rebound, common cold, hypertension, headache, weight gain, nausea/diarrhea, dysaesthesia, skin infection, dyslipidaemia, elevation of liver enzymes, elevation of creatinine
Haeck et al. (2011) [15]	Netherlands	November 2005 to November 2007	RCT	Adults (≥18 years)	Severe	26	Start CsA 5 mg/kg/day for 6 weeks and reduced to 3 mg/kg/day for 30 weeks	24	Start CsA 5 mg/kg/day for 6 weeks and changed to MMF 1440 mg/day for 30 weeks	Nausea, altered defecation pattern, headache, fatigue, paraesthesia, muscle ache, infections, flu, hypertrichosis, gum hyperplasia, lower leg edema, creatinine increase, anemia, leucopenia, thrombopenia, liver enzyme increase, cholesterol increase, magnesium increase, hypertension
El-Khalawany et al. (2013) [16]	Egypt	Not described	RCT	Children (8–14 years)	Severe	20	CsA 2.5 mg/kg/day for 12 weeks	20	MTX 7.5 mg/week for 12 weeeks	GI disorder, hematologic disorder, elevated ESR, abnormal liver function tests, abnormal renal function tests, fever, fatigure, headache, hypertension, flu-like symptoms
Goujon et al. (2018) [13]	France	December 2008 to March 2012	RCT	Adults (≥18 years)	Moderate to severe	43	CsA 2.5 mg/kg/day for 8 weeks and increased to 5 mg/kg/day for 16 weeks in poor response cases	50	MTX 15 mg/week for 8 weeeks and increased to 25 mg/week for 16 weeks in poor response cases	Infections, pain/nervous system disorder, GI disorder, hypertension, skin disease

Abbreviations are as follows: AD, Atopic dermatitis; CsA, cyclosporine A; GI, gastrointestinal; flu, influenza; MMF, mycophenolate mofetil; MTX, methotrexate; RCT, randomized clinical trial.

**Table 2 jcm-12-01390-t002:** Data summary of studies included in the meta-analysis.

Treatment	Number of Studies	Number of Observations	Estimate	95% CI	*p*-Value
Atopic Dermatitis Severity Score (SMD)					
High-dose CsA	2	155	−3.229	(−11.534, −5.076)	<0.001
MTX	3	187	1.995	(−5.435, 9.424)	0.599
MMF	1	50	−11.193	(−13.540, −8.845)	<0.001
PRD	1	38	0.096	(−0.543, 0.736)	0.768
Overall	7	430	−1.617	(−6.468, 3.234)	0.5135
Adverse Events (IRR)					
High-dose CsA	1	106	1.2	(0.663, 2.172)	0.547
MTX	2	137	0.64	(0.381, 1.076)	0.092
MMF	1	50	0.634	(0.444, 0.905)	0.012
PRD	1	38	0.787	(0.494, 1.253)	0.313
Overall	5	331	0.725	(0.564, 0.932)	0.012

Abbreviations are as follows: CsA, cyclosporine A; CI, confidence interval; IRR, incidence rate ratio; MMF, mycophenolate mofetil; MTX, methotrexate; PRD, prednisolone; SMD, standardized mean difference.

## Data Availability

The data used to support the findings of this study are included within the article.

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
