# Peer review of "Efficacy and Safety of Low-Dose Cyclosporine Relative to Immunomodulatory Drugs Used in Atopic Dermatitis: A Systematic Review and Meta-Analysis"

_jcm, 2023, doi:10.3390/jcm12041390_

Round 1
Reviewer 1 Report
Authors performed a meta-analysis of randomized placebo-control trials of treating atopic dermatitis (AD), comparing low-dose (<4 mg/kg) cyclosporin A (CsA) as control with other systemic treatments, including high-dose (≥4 mg/kg) CsA, methotrexate (MTX), mycophenolate mofetil (MMF) and prednisolone (PRD) as experimental treatments in the efficacy. They found that low-dose CsA was not inferior to high-dose CsA and other systemic treatments. Somehow, low-dose CsA showed significantly higher incidence of adverse events than the other systemic treatments. However, the incidence of serious adverse events of low-dose CsA was not significantly different from, and rather lower than that of the other systemic treatments.
Major problems:
1. The high-dose of CsA was reduced to low-dose after 6 weeks in Haeck’s study, whereas the treatment with CsA was started at a low dose and increased to a high-dose on Goujon’s study. The results of later phases should be carefully analyzed, or ideally excluded from the analysis, because they are largely biased in terms of subject selection.
2. As stated above, high-dose CsA was taken in both study by Schmitt, et al. and that by Haeck et al. If only results of Schmitt et al was taken in the analysis, it should be clearly described.
3. A meta-analysis of the effect and safety of CsA for atopic dermatitis until 2005 was reported by Schmitt et al covering lower levels of evidence than this study. Comparison of that result with this study should be described and discussed.
4. A possible reason for discrepancy between the occurrence of adverse events (AEs), higher in low-dose CsA, and serious AEs, lower in low-dose CsA, should be discussed.
5. Lastly, and most importantly, I feel it very odd to compare the efficacy and safety of low-dose CsA with those of high-dose CsA and other systemic treatments all together. It is much more reasonable to compare low-dose CsA and only high-dose CsA, or low-dose CsA and non-CsA systemic treatments. These two comparisons are largely different in many aspects. The former is essentially comparing doses of the same medication, whereas the latter is comparing mechanisms of action by different medications.
Reviewer 2 Report
Good and quite interesting paper
However, some point should be elucidated:
Table 1:
- haeck et al. In the control group has been reported CsA 5 mg/kg/die, but authors defined control group when the CsA dosage was lower than 4 mg/kg/die - explain this point
- goujon et al. In the control group has been reported CsA 2.5 mg/kg/die increased to 5 mg/kg/die (higher than 4 mg/kg/die) - explain this point also
Figure 4
the IRR of AEs was lower in the high-dose CsA and other systemic treatments group than in the low-dose CsA : authors should explain how they justify that AEs was lower in the high-dose CsA comparing with the low-dose CsA
In the limitations should be emphasized the limitation of the reliability of the results, since few studies were selected and there was great variability in dosages and interventions
Reviewer 3 Report
This is a systematic review and meta analysis about efficacy and safety of low dose CyA vs other systemic drugs
In the abstract you talk about a "children" population, while in the text you mention adult and children: not clear (line 23-24)
Line 32 "Our study may justify the use of low-dose CsA, rather than high-dose CsA and other 32 systemic immunomodulatory agents in moderate-to-severe AD" why? if you say (line 160-161) that "When comparing low-dose CsA with high-dose CsA, low-dose CsA showed no significant difference in incidence of AEs and serious AEs"
In the introduction you mentioned (line 43-) systemic immunomodulatory agents but I can't find jak inhibitors, and no reference regarding the new EDF guidelines on atopic eczema (JEADV 2022)
Line 50: effect-replace with efficacy
table 1: you didn't mention about the duration of the studies (you compared data at 8-12 weeks and "at the end" of the study without indicating the total lenght of the trials)
Line 102: what do you mean by "the number of considered AEs?"
Could you expand the paragraph regarding "adverse events and severe AEs" by desdcribing them?
line 180- "In general, it is recommended to start with a higher dose of 4 – 5 mg/kg/day to obtain a good initial result unless the patient is old or suffers from relevant concomitant diseases, reference 19: " The mean age of the 22 patients with a clinically relevant serum creatinine increase in our study was significantly higher than the mean age of patients without this increase in serum creatinine".
I think that you should expand the discussion: it is not well understood why low dose CyA has higher rates of AEs than high dose, and if yes, which kind of AEs and why.
Round 2
Reviewer 3 Report
Dear authors, thank you for your reply
I think you made appropriated changes to the article
It is an interesting article even if I think it should have been designed better
line 198: A previous study found that some patients may tolerate low-dose CsA: ?
Why did you included Hack's study if you then eliminate from your analysis
Line 212-213 An initial daily dose of 5 mg/kg/day, divided into two single doses, is recommended. A dose reduction of 0.5 – 1.0 mg/kg/day every 2 weeks is recommended, once clinical efficacy is reached: it is not clear if it's a general recommendation or you are talking about the guidelines cited, in particular you say the opposite in your paper. Better to reformulate this part
line 208: Nephrotoxic effects are more likely to occur if the daily CsA dose exceeds 5 mg/kg body weight: better to mention about the duration of therapy
Check for English language
Author Response
"Please see the attachment."
